# Fast-growing species shape the evolution of reef corals

Alexandre C. Siqueira [1,2 ✉], Wolfgang Kiessling [3] & David R. Bellwood [1,2]

Ecological interactions are ubiquitous on tropical coral reefs, where sessile organisms coexist in limited space. Within these high-diversity systems, reef-building scleractinian corals form an intricate interaction network. The role of biotic interactions among reef corals is well established on ecological timescales. However, its potential effect on macroevolutionary patterns remains unclear. By analysing the rich fossil record of Scleractinia, we show that reef coral biodiversity experienced marked evolutionary rate shifts in the last 3 million years, possibly driven by biotic interactions. Our models suggest that there was an overwhelming effect of staghorn corals (family Acroporidae) on the fossil diversity trajectories of other coral groups. Staghorn corals showed an unparalleled spike in diversification during the Pleistocene. But surprisingly, their expansion was linked with increases in both extinction and speciation rates in other coral families, driving a nine-fold increase in lineage turnover. These results reveal a double-edged effect of diversity dependency on reef evolution. Given their fast growth, staghorn corals may have increased extinction rates via competitive interactions, while promoting speciation through their role as ecosystem engineers. This suggests that recent widespread human-mediated reductions in staghorn coral cover, may be disrupting the key macroevolutionary processes that established modern coral reef ecosystems.

[1] Research Hub for Coral Reef Ecosystem Functions, College of Science and Engineering, James Cook University, Townsville, QLD 4811, Australia. [2] ARC Centre of Excellence for Coral Reef Studies, James Cook University, Townsville, QLD 4811, Australia. [3] GeoZentrum Nordbayern, Friedrich-Alexander University Erlangen - Nürnberg (FAU), Erlangen 91054, Germany. ✉email: alexandre.siqueira@my.jcu.edu.au

Biological diversity stems from the interplay between biotic and abiotic controls, operating across temporal and spatial scales[1]. Among high-diversity systems, coral reefs stand out for harbouring the majority of marine species, despite occupying relatively small areas[2]. Coral reef biodiversity has been shaped by major changes in climate, ocean chemistry, sea level and nutrient dynamics through geological time[3]. However, while some of these abiotic factors influenced large-scale patterns of reef biodiversity (through mass-extinctions[4], for example), biotic controls have also been suggested as important determinants of the temporal waxing and waning of reefs[3]. Specifically, competition for space has been hypothesized as a fundamental driver of reef evolution[3].

On short ecological timescales, interspecific competition is regarded as one of the key mechanisms shaping the composition of coral reef communities[5–8]. As the namesake of these ecosystems, reef-building corals (Order Scleractinia) are major contributors to habitat formation and are often one of the most abundant benthic sessile invertebrates on present-day reefs[9]. Given the limited space available on shallow marine reefs, corals can compete through direct interference and overgrowth, or through indirect pre-emption of space for larval settlers[5–8]. Alongside tropical rainforests, coral reefs have been used to typify biological systems moulded by interspecific competition, including a foundational work in theoretical ecology[5]. Yet, it is not known whether competition for space or other biological interactions can scale up to determine large-scale macroevolutionary patterns on coral reefs.

Here we show that the diversification of Acroporidae (commonly known as staghorn corals) is associated with a major disruption in coral evolutionary patterns, suggesting strong diversity-dependent effects (i.e., when diversification in one lineage impacts the evolution of others[10]). Using recently developed Bayesian modelling techniques[11,12], we assessed the diversification dynamics of scleractinian coral fossil lineages at the species level. By focusing on the Cenozoic, we: (a) describe the major temporal changes in evolutionary rates that determined the formation of present-day species richness patterns among scleractinian coral families; (b) explore fossil diversity trajectories of major extant reef-building coral families; and (c) estimate the potential effects of environmental and biological drivers on speciation and extinction rates across reef coral families. Our results reveal a unique mechanism through which biological interactions may shape the evolution of high-diversity systems.

## Results and discussion

**Recent shifts in diversification.** We detected three major episodes of extinction in scleractinian corals during the last 70 million years [Myr]. The first is the mass-extinction event at the Cretaceous–Paleogene boundary (Fig. 1a). This event is known for having affected corals only moderately relative to other marine invertebrates[13], however, it was still significant at the species level for Scleractinia. The second, at the Eocene–Oligocene transition (Fig. 1a), coincided with a major cooling event that transformed the world into an icehouse after the hot climate of the Eocene[14]. Although this transition is not classically recognized as a mass-extinction[15], for reef-building corals this cryptic extinction event[16] seems to have substantially impacted diversity. Third, we found that coral extinction also increased sharply in the Pliocene–Pleistocene. Remarkably, this extinction also coincided with a striking increase in speciation rates (Fig. 1a).

Recent shifts in coral diversification have been noted previously[17], but their causes remain elusive. Given that the Pliocene-Pleistocene is not associated with a mass-extinction event, we ask: could these shifts have been driven by biotic

interactions? By running the same models across extant reef-building coral families, we found a remarkable spike in speciation rates of Acroporidae in the last 3 Myr (Fig. 1b). But notably the speciation spike was not mirrored by extinction (Fig. 1b). Consequently, we found very high net diversification rates in the Acroporidae close to the present (Fig. 1b), even when compared to the recent diversification of all scleractinians (Fig. 1a). This suggests that acroporids had a markedly different diversity trajectory to most other scleractinian corals, and if diversity dependency was involved, they may have been the winners.

**Diversity trajectories and dependency.** The identified shifts in evolutionary rates point to the Pliocene–Pleistocene as a key period for the formation of modern reef-building coral species richness. Therefore, to further explore this accumulation of diversity, we focused on species that are classified as reef-associated and examined the evolution of coral families that are: (i) abundant on present-day reefs (Acroporidae, Agariciidae, Merulinidae, Mussidae, Pocilloporidae, and Poritidae)[18]; and (ii) have a high number of occurrences in the fossil record. We estimated fossil diversity trajectories through time[12] for each of these six families separately. These models revealed major temporal fluctuations in the number of lineages, particularly in the Miocene (Fig. 2a). But more importantly, they show that for most of its evolutionary history, the Acroporidae was not the most diverse family of corals. It was only very recently, in the Pleistocene, that acroporids became the most species-rich group of reef-building corals (Fig. 2a). Although their diversity almost doubled from the late Miocene to the Pliocene (8–3 Million years ago [Ma]; Fig. 2a), the rate of change was substantially higher in the Pleistocene (Fig. 2b).

To assess the main factors driving these macroevolutionary trends, we ran models that estimated the correlation between diversity trajectories and both environmental and biological predictors[11] (see Methods). Results from these models show that the environmental factors tested—paleotemperature, sea level and rate of sea-level change—were generally less well correlated with fossil trajectories than biological predictors (i.e., the diversity of other coral taxa). The only detectable potential abiotic effects were: (i) a slight increase in speciation rates in the Acroporidae that may have been associated with the rate of sea-level change, and (ii) a negative correlation between paleotemperatures and extinction rates (i.e., higher temperatures associated with less extinction) in the Pocilloporidae (Supplementary Fig. 1). Our models of cross-clade effects, on the other hand, suggest that diversity dependence was widespread throughout reef coral evolution (Fig. 3). But the directionality of effects was not taxonomically homogeneous. The Acroporidae stand out as the only family that may have affected both speciation and extinction in all families combined (Fig. 3a, b; central node). This overwhelming effect had contrasting outcomes; while acroporids promoted extinction in other coral lineages (Fig. 3b; central node), they also seem to have promoted speciation (Fig. 3a; central node). These results were consistent when we performed the same analysis, selecting only sites where Acroporidae species co-occur with the other families (Supplementary Fig. 2), and on Indo-Pacific species only (Supplementary Fig. 3). Family pairwise analyses also showed similar results, with Acroporidae potentially enhancing speciation and extinction in five and three out of six families, respectively (Fig. 3a, b; external nodes). No other family had this many effect links with such a strong overall intensity; and this is despite the fact that the Acroporidae was not the most abundant nor the most diverse coral family before the Pleistocene (i.e., results were not driven by prevalence in the palaeontological

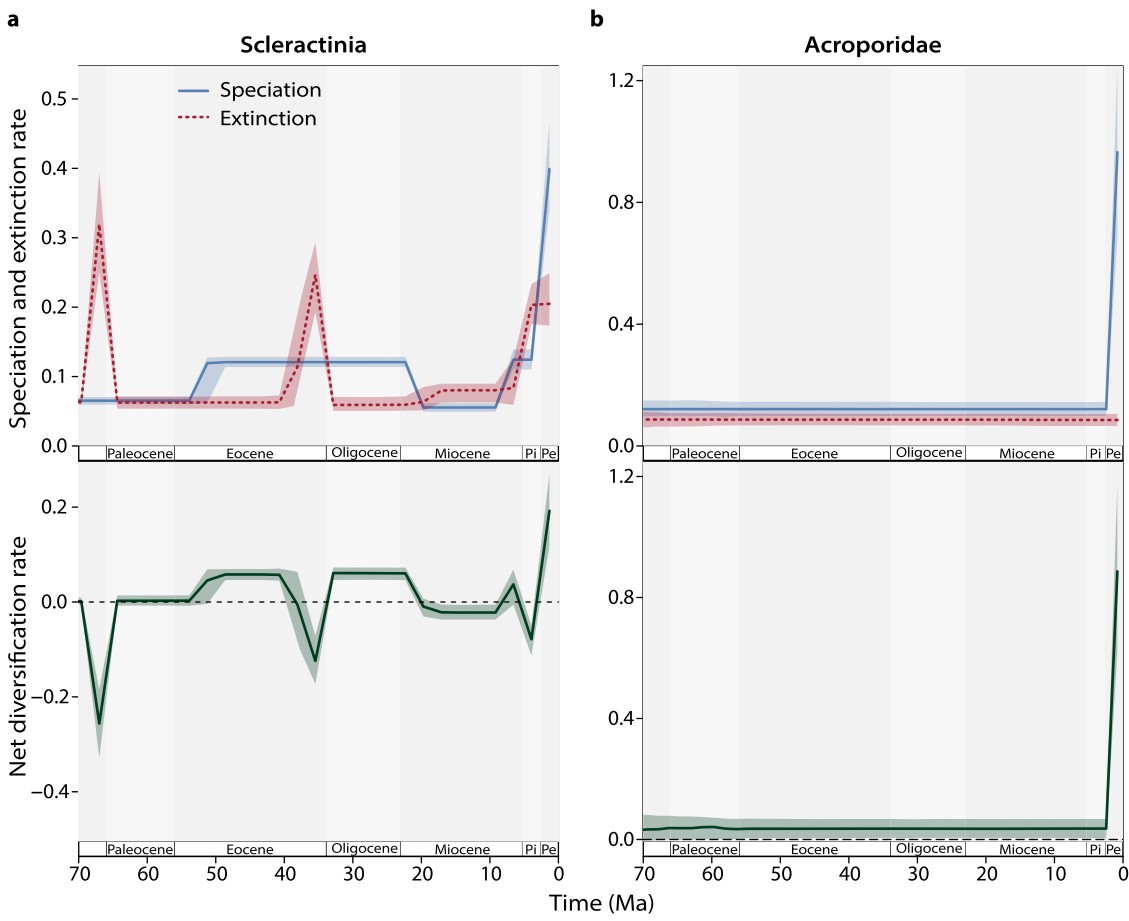

**Fig. 1 Evolutionary rates through time in scleractinian corals.** Rates of speciation (blue) and extinction (red dashed) estimated for fossil lineages of Scleractinia (**a**; $n = 4235$) and Acroporidae (**b**; $n = 165$). The analyses incorporate all the fossils recorded in the respective groups, although here we only show the last 70 million years. The resulting net diversification rates (speciation minus extinction) are shown at the bottom panels. Solid and dashed lines represent the median rates, while coloured shadings represent 95% posterior credibility intervals. Pi—Pliocene; Pe—Pleistocene. Source data are provided as a Source Data file.

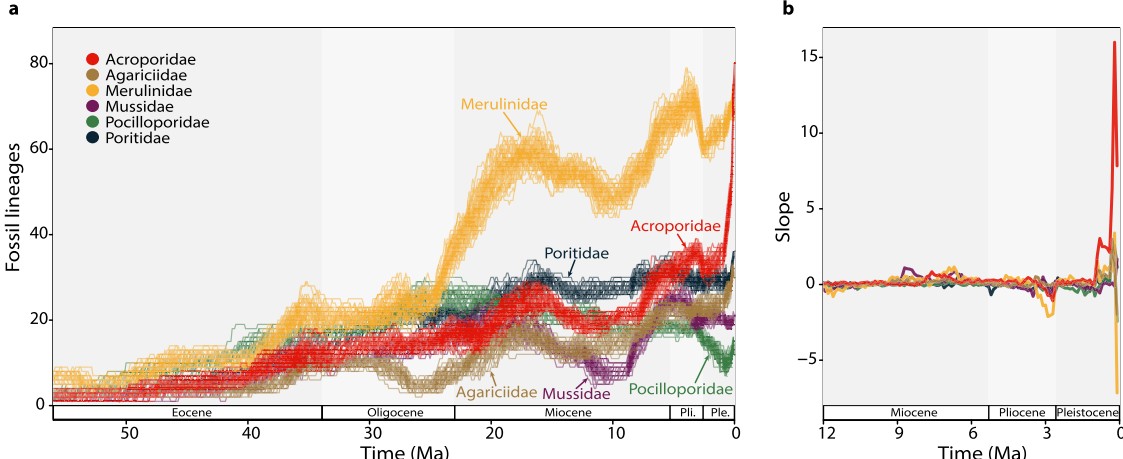

**Fig. 2 Fossil lineages through time in six extant reef coral families. a** Reconstruction of fossil diversity trajectories estimated through a time-variable Poisson process of preservation. Models were run independently in each family and were replicated fifty times (individual lines) to incorporate uncertainty around the age of the fossil occurrences. **b** Mean rate of diversity change (slope in species per 0.1 Myr) in the last twelve million years for the same coral families, calculated as the difference in diversity between subsequent time points (see Methods). Acroporidae—red; Agariciidae—brown; Merulinidae—yellow; Mussidae—purple; Pocilloporidae—green; Poritidae—dark blue. Source data are provided as a Source Data file.

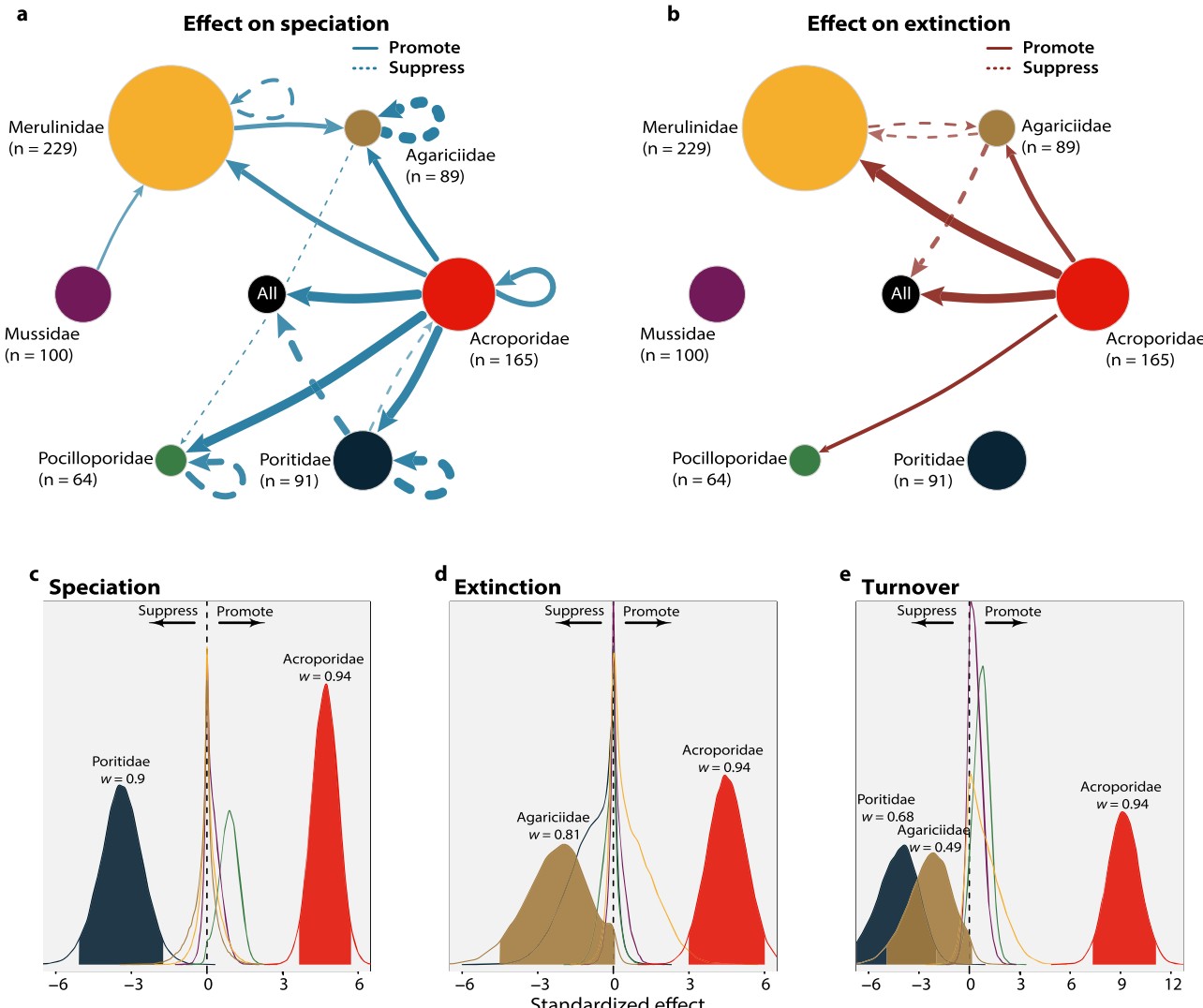

**Fig. 3 The overwhelming cross-clade effect of Acroporidae on the diversity of reef corals.** A network depicts the estimated cross-clade influence on rates of speciation (**a**) and extinction (**b**). Arrows represent the directionality of interaction effects, while arrow widths are proportional to the estimated median effect. Continuous or dashed lines indicate promoting or suppressing effects, respectively. Only interactions with a strong signal in the model were included ($w > 0.7$; see Methods). Loops represent self-diversity dependency; i.e., when confamilial species influence their own diversity trajectories. The size of each external node is proportional to the number of fossil occurrences in each family, and the number of fossil species in each family is shown in parenthesis ($n$). The central node represents all families combined, for which we also show the posterior distribution of effects on speciation (**c**), extinction (**d**) and lineage turnover (**e**; effect on speciation plus effect on extinction). Filled densities represent effects estimated with a strong signal ($w > 0.7$). The median shrinkage weights ($w$) are also shown for these families. Acroporidae—red; Agariciidae—brown; Merulinidae—yellow; Mussidae—purple; Pocilloporidae— green; Poritidae—dark blue. Source data are provided as a Source Data file.

record). Moreover, while it is reasonable to expect strong self-diversity dependency (i.e., species within the same family influencing their own diversity trajectories[19]), given that confamilial species generally share more similar ecologies, this does not seem to be the case for reef corals. We did find self-suppressing effects on speciation in four families (Fig. 3a). However, these effects are negligible when compared to the strong cross-clade influence of Acroporidae on overall speciation (and its own self-promoting effect; Fig. 3a) and extinction.

A key feature of the model used herein is that it provides a robust measure to distinguish between noise and signal in the diversity-dependent effects (shrinkage weight [$w$]—where 0 represents noise and 1 represents signal[11]; see Methods). This measure reinforces our main results: the median shrinkage effect weight was substantially higher in Acroporidae ($w_{Acro} = 0.94$; Supplementary Table 1) than in other families. More importantly,

the model is blind to species ecological attributes, yet it detected biologically meaningful results. The only family that enhanced overall speciation (Acroporidae; Fig. 3c) is mostly composed of fast-growing branching species[9] (Supplementary Fig. 4) that are major ecosystem engineers that contribute to habitat complexity[20]. On the other hand, the massive corals of the family Poritidae, that generally reduce small-scale habitat heterogeneity[8], were found to suppress speciation (Fig. 3c). Today, with their fast growth, acroporids tend to be competitively superior under intermediate disturbance regimes[7,21]. Competition may, therefore, at least in part, explain why they are the only group to promote extinction in the models (Fig. 3d). The buffering effect of Agariciidae against extinction (Fig. 3d) is the only result that defies a straightforward ecological explanation and should be investigated further. But it is in the sum of these effects that the Acroporidae prevail. The models suggest that

acroporids contributed to a nine-fold increase in overall lineage turnover throughout the evolution of reef corals (Fig. 3e).

Our results provide compelling evidence for the presence and potential strength of biological interactions in the evolution of modern coral reefs[3]. They also suggest that rapid sea-level variation[22,23] may only be a partial explanation for recent changes in the relative prevalence of reef coral families. Earlier biological interactions may have shaped both diversity patterns and subsequent prevalence. At first glance, it seems paradoxical that one coral family can simultaneously enhance overall rates of speciation and extinction. However, it is important to note that these effects could vary: 1—between lineages within the same family (i.e., some species might go extinct while others originate during the same time period); and 2—across time (i.e., effects are estimated for the entire duration of the temporal overlap between lineages). Regardless, these seemingly contrasting effects are easily explained by the unique ecology of acroporids. Species within this family are the fastest calcifying corals on reefs[24], dominating the benthos and leaving few opportunities for other groups to grow when levels of disturbance are infrequent, but intense[21]. For example, in one of the few long-term ecological assessments of competition on reefs, branching *Acropora* species were found to be competitively superior and, subsequently, reduced coral diversity[7]. Thus, it seems likely that the weedy nature of staghorn corals pushed other coral species to extinction during low sea-level stands in the Pleistocene, a critical time when habitat area was reduced[25,26]. Despite this inferred effect on extinction, the high accretion rates and reef-building capacity of acroporids[24] may have had a double-edged effect on reef coral biodiversity by also promoting speciation in other scleractinian families. As recently demonstrated, acroporids are major contributors to the fractal dimension of reef complexity[20], which not only promotes niche differentiation, but may also facilitate coral larvae settlement by creating fine-scale structures[27]. Since staghorn coral growth forms are more likely to be damaged by intense hydrodynamic disturbances[28], they might also have facilitated speciation of opportunistic species that flourish as new space becomes available[29]. In addition, by tracking sea-level changes, acroporids increased the heterogeneity of reef types in the Pleistocene[30], which likely favoured habitat specialization in other coral groups.

The fast and diversified growth forms of acroporids[9] were, therefore, key to the development of present-day coral reefs. Ironically, however, their life strategy seems to be most affected by recent human impacts including climate change[31–33]. Although acroporids thrived with the moderate disturbance levels of the Pleistocene, the current rate of change is severe[33] and reef-building corals may struggle to adapt. Framework-building and habitat generation by acroporid corals has already declined globally[34], with demonstrated negative consequences for reef-associated faunas[35]. We show that the ecological attributes of acroporids are not only important for supporting modern reef species, but were also the likely promoters of their diversity. This provides a bitter lesson for the current trajectory of biodiversity on coral reefs: we may be causing irreversible changes in the fundamental evolutionary mechanisms that created these exceptionally rich ecosystems in the first place.

## Methods

**Fossil data**. We downloaded all fossil occurrences recorded for the order Scleractinia at the species level from the Paleobiology Database (PBDB – paleobiodb.org; accessed on 3 August 2021). This is the most comprehensive repository for palaeontological data in reef corals to date. Due to the nature of the data, no ethics approval was required. To minimize identification issues, we excluded taxa with uncertain generic and species assignments (i.e., classified as aff. and cf.) and only selected species that had accepted names. We also selected the variables *classification* and *palaeoenvironment* from the output options to facilitate taxonomic and environmental filters applied in downstream analyses. The full dataset

consisted of 24,011 occurrences across 4235 species, spanning over 250 Myr of coral evolution from the Triassic to the present. Although our focus here lies on the Cenozoic, we used the complete fossil dataset (i.e., including all of the occurrences) to have estimates of the diversification dynamics in scleractinian corals throughout the whole timespan of their evolution.

**Evolutionary rates**. With the full palaeontological dataset, we estimated evolutionary rates through time in scleractinian corals using the Bayesian framework of the program PyRate (v3.0)[12,36,37]. This program uses fossil occurrence data to calculate the temporal variation in rates of preservation, speciation and extinction, while incorporating multiple sources of uncertainty[12]. At its core implementation, PyRate jointly estimates the times of origination ($Ts$) and extinction ($Te$) for each fossil lineage; the fossilization and sampling parameters that determine preservation rates ($q$); and the overall rates of speciation ($\lambda$) and extinction ($\mu$) through time[36]. Recently, the program has been upgraded to include a reversible jump Markov Chain Monte Carlo (rjMCMC) algorithm to estimate diversification rate heterogeneity, which provides more accurate and precise estimates than other commonly used methods[12]. Therefore, despite the inherent bias of the fossil record (i.e., estimates are conditioned on sampled lineages), PyRate is a robust method to quantify speciation and extinction rates, and their respective temporal shifts, from fossil occurrence data.

Extant taxa can also be included in the PyRate framework as long as they are also represented in the fossil record. This is done to extend the fossil geologic ranges to the recent times. Hence, the first step in our analysis was to identify which species in our dataset is still alive at the present. To do this, we matched the accepted species names in the PBDB dataset with those from the extant species dataset of Huang et al.[38]. Subsequently, we split our dataset into eleven independent subsets, with the goal of keeping each subset with an equal number of species. Each data subset included a random selection of species with their respective occurrences, which was enough to calculate $Ts$ and $Te$ (see below). This was done to avoid convergence issues, given the large size of our dataset and the consequent complexity of the model[37]. For each of our subsets, we generated fifty replicates by resampling the fossil ages from their temporal ranges to account for the uncertainty associated with the age of occurrences. We then used the maximum-likelihood test in PyRate to compare between three models of fossil preservation[12]: the homogeneous Poisson process (HPP; $q$ is constant through time); the nonhomogeneous Poisson process (NHPP; $q$ varies throughout the lifespan of a species); and the time-variable Poisson process (TPP; $q$ varies across geological epochs). The latter model (TPP) was selected across all of our data subsets (Supplementary Table 2).

After selecting the preservation model, we first focused on assessing the estimates of times of origination and extinction in each data subset, rather than using the full dataset to jointly estimate all parameters at once as in the original implementation of PyRate. This further reduced the complexity of the model and allowed for more precise parameter estimates. For each replicate in all of our data subsets, we approximated the posterior distribution of $Ts$ and $Te$ through a 50 million generation run of the rjMCMC algorithm under the TPP, sampling parameters every 40 thousand iterations. At the end of each run, we discarded 20% of the samples as burn-in and assessed chain convergence through the effective sample sizes of posterior parameter estimates, using the software Tracer[39] (v1.7.1).

From the results of this first set of models, we extracted the median estimates of $Ts$ and $Te$ across replicates, and we merged the estimates from the eleven independent data subsets. This merged data frame contained estimated times of origination and extinction for all coral lineages within our fossil dataset. We then used this merged $Ts$ and $Te$ data frame as input for another rjMCMC chain to finally estimate overall $\lambda$ and $\mu$ through time, by applying the option -d in PyRate. In this option, $Ts$ and $Te$ for all fossil lineages are given as fixed values and, therefore, are not estimated by the model. The chain for this model was run for 100 million generations, sampling parameters at every 40 thousand iterations. Once again, we excluded 20% of the initial samples as burn-in and checked model convergence using Tracer. Finally, we calculated net diversification rates through time by subtracting the post burn-in samples of $\mu$ from $\lambda$.

To explore the taxonomic idiosyncrasies in the evolutionary rates of reef corals, we selected the most abundant families on present-day coral reefs in terms of the number of colonies per area[18] (Acroporidae, Agariciidae, Merulinidae, Mussidae, Pocilloporidae, and Poritidae). Altogether, species within these families account for ~40% of the total extant diversity in Scleractinia. These families also account for most of the occurrences in the PBDB fossil dataset (excluding extinct families, which are generally older and had little temporal overlap with extant ones): Acroporidae (1457 occ. in 165 spp.); Agariciidae (722 occ. in 89 spp.); Merulinidae (2464 occ. in 229 spp.); Mussidae (1146 occ. in 100 spp.); Pocilloporidae (615 occ. in 64 spp.); and Poritidae (1149 occ. in 91 spp.). Therefore, from our full dataset, we selected six independent ones encompassing all species in each of the selected families. We also selected only species that are classified as reef-associated within these families, since we were specifically interested in these environments. This selection had a negligible effect on the size of the individual datasets, given that the vast majority of fossil species within these families are reef-associated. In each family, we followed the same modelling steps described above to estimate $\mu$ and $\lambda$, and diversity trajectories. However, this time it was not necessary to split the datasets into subsets, given that each family has far less occurrences than the full dataset. We started by comparing

models of preservation, which showed the TPP as the best supported for all families (Supplementary Table 3). Then we created fifty replicates by resampling fossil ages to accommodate the uncertainty associated with the time of occurrences. For each replicate, we ran the rjMCMC algorithm for 50 million generations under the TPP model, with a sampling frequency of 40 thousand iterations. We discarded initial 20% of the samples as burn-in, and assessed convergence through Tracer. We then combined all replicates, resampling 100 random samples from each replicate to assess the estimates of $\mu$ and $\lambda$ through time for each family. Finally, we extracted diversity trajectories in each family for all of the replicates by applying the -ltt option in PyRate, which generates a table with estimated range-through diversity at every 0.1 Myr. From these trajectories, we calculated the mean difference in diversity (slope in species per 0.1 Myr) between subsequent time samples backwards from the present (i.e., diversity in time $t$ was subtracted from diversity in time $t$-1) using the *diff* function in R (v4.0.3).

As an alternative to PyRate, we also calculated the diversity dynamics of reef coral fossils using the R package divDyn[40], which combines a range of published methods for quantifying fossil diversification rates. Differently from PyRate, the metrics applied in divDyn require that the fossil occurrences are split into discrete time bins. Therefore, these metrics treat the origination and extinction rates as independent parameters in each bin, while PyRate is designed to detect rate heterogeneity through a continuous time setting[12]. Our goal here, however, was not to compare models but to assess the robustness of our rate patterns and diversity trajectories using alternative methods. We divided our dataset into one-million-year time bins to have enough temporal resolution for rate calculations. To account for the uncertainty in the assignment of fossil ages, we created 50 binned replicates by sampling the age of each occurrence from a random uniform distribution, with bounds defined by the age ranges provided in the PBDB dataset. We then used the *divDyn* function to calculate the per capita rates of origination and extinction through time (based on the rate equations by Foote[41]) for all scleractinians (Supplementary Fig. 5a) and for reef-associated acroporids alone (Supplementary Fig. 5b). We also used the same procedure to generate range-through diversity curves for each of the six families selected previously, to compare with the curves generated by PyRate (Fig. 2a). Although the rate results differed between the PyRate (Fig. 1) and the divDyn (Supplementary Fig. 5) approaches, the general patterns remained unchanged. Rates are more volatile through time in divDyn estimates, with larger confidence intervals, which is expected from the metrics applied in the package[12,42]. Yet, we found the same peaks in extinction for Scleractinia: at the Cretaceous-Paleogene and Eocene-Oligocene boundaries, and at the Pliocene-Pleistocene (Supplementary Fig. 5a). The recent peak in speciation in Acroporidae was also detected, although less strong (Supplementary Fig. 5b). Despite these slight differences in rate estimates, the diversity curves reconstructed through divDyn (Supplementary Fig. 6) mirrored almost exactly the ones found with PyRate (Fig. 2), demonstrating that the overall macroevolutionary trends described herein (Figs. 1 and 2) are robust to methodological choices.

**Diversity-dependent models.** To assess the effects of diversity dependency on the evolution of reef coral lineages, we implemented the Multivariate Birth-Death model (MBD)[11] within the PyRate framework. This method was first described as the Multiple Clade Diversity Dependence model (MCDD)[19], in which rates of speciation and extinction are modelled as having linear correlations with the diversity trajectories of other clades. At its original implementation, the MCDD was developed to assess the effects of negative interactions, where increasing species diversity in one group can suppress speciation rates and/or promote extinction in itself or in other ecologically similar clades[19]. However, the model also incorporates the possibility of positive interactions, where increasing diversity in one clade can correlate with enhanced rates of speciation or buffered extinction. Through further model developments[43], the MCDD was updated to also include a horseshoe prior[44] on the diversity-dependence parameters, which helped controlling for overparameterization and enhanced the power of the model to recover true effects[43]. More recently, this model took its current form as the MBD[11], with the additional possibilities of including environmental correlates and setting exponential, rather than just linear, correlations.

We first applied the MBD to estimate the diversity-dependent effects of individual extant coral families (i.e., the ones selected in the previous analysis; see Evolutionary rates) in their combined diversity trajectories. From the rjMCMC model results for individual families, we extracted estimates of $Ts$ and $Te$ in each of the fifty replicates and merged them across families. This merged dataset with fifty replicates of $Ts$ and $Te$ was then used as input for the MBD model, where we set the relative diversity trajectories of each individual family as predictors. We also included three key environmental predictors—paleotemperature, sea level and rate of sea-level change—to assess their influence in overall evolutionary rates. The paleotemperature data was obtained from Westerhold et al.[45], and consists of global mean temperature estimates for the last 66 million years, averaged across 0.1 Myr time bins. Eustatic sea-level data was downloaded from Miller et al.[46], and contains estimates of sea level for the last 100 million years in comparison to present-day levels, also split in ~0.1 Myr time bins. With this dataset, we calculated the average rate of sea-level change per million years, as measured from the absolute difference between subsequent sea-level values backwards in time (i.e., sea level in time $t$ was subtracted from sea level in time $t$-1). These environmental factors were rescaled between 0 and 1 to maintain all predictors on the same relative scale.

Under our MBD model, the speciation and extinction rates of all families combined could change through time and through correlations with the relative diversity of individual families or environmental factors. The strength and directionality (positive or negative) of the correlations are also jointly estimated for each predictor within the model[11]. We ran both linear and exponential correlation models (see formulas in Lehtonen et al.[11]) in each of our fifty replicates for 25 million generations, sampling parameters at every 25 thousand iterations. We then compared the linear and exponential models through the posterior harmonic means of their log likelihoods, which supported the exponential one as having a better fit. From the posterior estimates, we summarized the speciation (Fig. 3c) and extinction (Fig. 3d) correlation parameters (i.e., the strength of the effect) by calculating their median and 95% Highest Posterior Density (HPD) interval across replicates. Finally, we also summarized the effect of families on lineage turnover (Fig. 3e), which we conceptualize as the sum of the effects on speciation and extinction.

The MBD model also provides posterior samples of the weight of the correlation parameters, which is estimated through the horseshoe prior[11]. In essence, this prior is able to reliably distinguish correlation parameters that should be considered noise from those that represent a true signal in the data[11]. The parameterization of the horseshoe prior contains local and global Bayesian shrinkage parameters[44] from which shrinkage weights ($w$) can be calculated (see formulas in Lehtonen et al.[11]). These shrinkage weights associated with each correlation parameter in the MBD model vary between 0 and 1, with values closer to 0 representing noise and values closer to 1 representing a true signal. Through simulations, it has been shown that values of $w > 0.5$ indicate that the correlation parameter in question significantly differs from the background noise, being the correlation positive or negative[43]. However, as a conservative way to infer the weight of correlation parameters, here we use a value of $w > 0.7$ to detect significance. This value was calculated for each diversity-dependence parameter (speciation, extinction and turnover) from the median values drawn from the model posteriors.

The spatial distribution of reef-associated taxa varied considerably throughout the Cenozoic, with biodiversity hotspots moving halfway across the globe[47]. Therefore, the best way to capture this dynamic biogeographic history in reef corals is by analysing global diversity patterns like we did in our main MBD model. However, to assess the robustness of our diversity-dependent results against the influence of geographic scale and site co-occurrences, we repeated all the modelling steps described above with two data subsets. First, we selected only fossil species that have occurrences in the Indo-Pacific Ocean (i.e., 30°W–180°W) within the six families. Second, we excluded sites in which the Acroporidae did not co-occur with the other families. In each of these data subsets, we calculated diversity trajectories and used them as predictors in a separate MBD model. These models had a merged dataset of $Ts$ and $Te$ of all species included in each case (Indo-Pacific and co-occurrences) as a response variable.

Finally, we followed the same modelling procedures described above to investigate the diversity-dependent effects in family pairwise analyses. We applied the MBD model to assess the effects of all other families in each individual family at a time, while also estimating correlations with the key environmental predictors. From the rjMCMC model results for individual families, we extracted the fifty replicates of estimated $Ts$ and $Te$. Each replicate was then used as input for an MBD run using the relative diversity trajectories of each other individual family as predictors, along with the environmental variables. Once again, we ran 25 million generations of the MBD, with a sampling frequency of 25 thousand, using both linear and exponential correlation models in each age replicate. For all families, we found that the exponential model had a better fit. We then summarized the correlation parameters and the shrinkage weights (Supplementary Fig. 7) derived from the exponential models per family by calculating the median and 95% confidence intervals across replicates.

**Reporting summary**. Further information on research design is available in the Nature Research Reporting Summary linked to this article.

## Data availability
The datasets generated and/or analyzed during the current study are available at the Zenodo repository (https://doi.org/10.5281/zenodo.6413373)[48]. There are no restrictions on data availability. All palaeontological data were collected from the publicly available Paleobiology Database (PBDB – paleobiodb.org; accessed on 3 August 2021). Source data are provided with this paper.

## Code availability
The PyRate program (v3.0) is written in Python (v3.) and is available at github.com/dsilvestro/PyRate. Tracer (v1.7.1) is available at github.com/beast-dev/tracer/releases/tag/v1.7.1. The R (v4.0.3) packages used were: tidyverse (version 1.3.1); scales (version 1.1.1); igraph (version 1.2.6); HDInterval (version 0.2.2); and shape (version 1.4.6). The codes used during the current study are available at the Zenodo repository (https://doi.org/10.5281/zenodo.6413373)[48].

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

## Acknowledgements

We are grateful to the PBDB team for their constant effort in maintaining their database and for making it freely available. We also thank K Quigley and S Tebbett for insightful comments, and D Silvestro for assistance with PyRate. Funding was provided by the Australian Research Council (DRB – LF190100062) and the Deutsche Forschungsgemeinschaft (WK – DFG – Ki 806/17-1). This is Paleobiology Database official publication number #424.

## Author contributions

A.C.S. and D.R.B. conceived the study. W.K. curated the data. A.C.S. performed the analyses and wrote the first draft of the manuscript. D.R.B. and W.K. contributed substantially to revisions.

## Competing interests

The authors declare no competing interests.
