## [Peer Review File · Nature Communications]

Fast-growing species shape the evolution of reef coralsREVIEWER COMMENTS

Reviewer #1 (Remarks to the Author):

This paper explores the importance of competition among reef corals over macroevolutionary timescales using broad fossil data and recently developed methods to assess diversity trajectories and their dependencies. The paper is particularly interesting because the role of competition is widely studied and well understood in ecological context, but its importance in shaping diversity patterns over evolutionary timescales have not been investigated before. Thus, this is significant and novel contribution to understand the evolution of reef systems and more generally, in understanding the role of biotic interactions in speciation-extinction dynamics.

The fossil data naturally are problematic. Current paper is based on extensive dataset, although it remains unclear what part of the full dataset (>24000 occurrences, >4000 species) were actually used in the final analyses (Cenozoic members of six selected families). Obviously full data were used to infer the preservation model. But it would still be good to know the number of species in the selected families and much they cover of the total standing reef coral diversity, especially because it is said that these families account for most of the occurrences in the dataset excluding extinct families. This immediately raises the question of the role the now extinct families may have played. Presumably they were driven extinct by the competition, if the results obtained with extant species would hold, and this could certainly change the results to some degree. It would have been nice to explore them as well, or better justify their exclusion. Two independent methods are used to estimate the diversification rates through time, and they seem to support similar conclusions.

The results are striking: the competitionally superior staghorn corals increase taxonomic turnover in other families, that is, both speciation and extinction. The authors are able to provide an explanation to this double-edged effect, by relating it to the capacity of staghorn corals to rapidly create complex reefs and hence, new niches for specialist species. I was wondering if it would be possible to backup this reasonable assumption with referencing some contemporary data?

The methods are as sound as any available method can be in this admittedly somewhat speculative field of research. I also do not see any such flaws in the methodology and interpretation that could have been avoided by using any other approach.

Reviewer #2 (Remarks to the Author):

Siqueira et al. show compelling results on the importance of the diversification of a family of reef-building corals impacting speciation and extinction of other coral families. They combine the fossil record with speciation and extinction rate analyses, lineage through time plots, and cross-clade models to show the importance of the family Acroporidae in influencing the evolutionary trajectories of other reef-building coral families. In general, this paper is exciting, well-written, and the results and methods are robust. I just have some minor criticisms with interpretation of the results and would like to see these further addressed.

1. How are you sure that it is competition is the driving force of extinction? You mention fast growth of acroporids likely promote their diversification, and this makes sense, but are there other processes that would promote the extinction/net-diversification of other reef building families? (i.e., environment is not conducive, other families are slow growing, etc). Also, it was not clear that you accounted for whether acroporidae co-occurred in the same location as other corals. If so, then I would be convinced that competition is important. Since you downloaded the fossil record database for your analyses, how did you account for whether the coral families co-occurred in the same locales? In other words, how could Acroporidae in the IndoPacific outcompete Pocilloporidae in the tropical eastern Pacific?

2. The cross clade network analyses are really cool. However, it is not clear how one family can simultaneously influence the speciation and extinction of another family. For example, in Figure 3a,

Acroporidae strongly promotes speciation of Pocilloporidae and you show in Fig 3b that it also strongly promotes the extinction of the same family. Your explanation in the text on lines 170-192 is interesting, and I understand where Acroporidae might impact speciation of one family and extinction of another, but I don't understand how it could influence both in one family. Unless it did this at different times? Also, could you run these analyses on net diversification rates? Perhaps that would provide additional insight.

Overall, nice work and a great read.

Reviewer #3 (Remarks to the Author):

Review for Fast-growing species shape the evolution of reef corals

In this contribution, the authors use data from the paleobiology to (lines 61-64)

- a) Described the major changes that determined the formation of present-day reef coral assemblages
- b) Estimate the effects of diversity dependency across reef-building coral families and
- c) Determine the relative importance of competition throughout the evolution of these families

In the abstract they conclude that (lines 24-25) "reef coral biodiversity experienced marked evolution rates shifts in the last three million years, likely driven by interspecific competition."

Much as I enjoyed learning something about corals from this ms, I do not think that the authors have shown c) or can draw their conclusion stated in the paragraph above (copied from their abstract). I think a) needs some revision and b) is conditioned on assumptions that need to be clearer (see below).

I will try to explain why in the following. In addition, I will try to clear on where the authors need to clarify their methods

Why are the conclusions problematic:

Basically, the authors used PyRate to estimate speciation and extinction rates from fossil data (Fig. 1), and the consequence of these rates (Fig. 2) i.e. the number of lineages given these rates (estimated within the same Bayesian model framework PyRate). These estimates show how coral richness changed over geologic time (given the PyRate framework) but I don't see how what the patterns estimated say anything about "coral assemblages" in a) which are more or less local (or at best regional) communities. The authors have estimated time-varying rates for corals (in their different groupings) but they have not gone beyond that as a) seems to indicate. Perhaps it is just a poor choice of words.

While using the MBD/MCDD framework, which basically models richness in one group to be linearly correlated with speciation and extinction rates of itself or with other groups, "diversity-dependence" is explored, but this analysis hardly allows one to conclude that it is ecological competition that leads to consequences in speciation and extinction rates. At best one might conclude that there is some signal between richness and rates (i.e. diversity-dependence is detected), but that is all. This is in part because, as the authors write in the paragraph starting with line 170, that there could be sea-level changes and other abiotic disturbances that could influence evolution. As an example, if sea-level was actually the primary driver of coral speciation, it could be the true underlying driver i.e. the solid arrows in Fig.3a, but because it was not modeled (not included in the analysis framework), one would conclude that it is Acroporidae that is driving speciation rates overall.

Without belaboring more specific examples, this means that how this ms is "sold" must be changed to reflect the underlying limitations of the analysis framework and data. If the authors really want to answer the questions they posed in the introduction, then other analyses must be carried out and other types of data collected.

In terms of the methods and description, I find it in general a bit wanting. E.g. the authors said they used the “complete fossil dataset” (line 216) to have “better estimates”. I don’t see any justification why this is so (nor have I come across that in the PyRate literature). In Lines 224-225, be careful to say that it PyRate estimates are conditioned on sampled lineages (i.e. all the estimates are conditioned on sampling at least once, which means that there will be some bias). In line 229 you cited Raup and Sepkoski for “commonly used” --- I am not sure if most people follow their estimation anymore? Please clarify. I would imagine Alroy (3 timer etc) or Foote (boundary crossers) are most commonly used? Line 232, the point of including extants is to have those fossil geologic ranges extend to the Recent right? Write that please if so, I think it is a bit misleading for people who don’t know how PyRate works. Line 236, why 11 subsets and how did you do it (“randomly”? but how exactly?) Why 11? Are your results sensitive to how this “division” was done? Seems completely arbitrary? Line 239: 10 replicates is much less than the 100 recommended in PyRate. I think that it is important to account for age uncertainty, so I could recommend at more replicates here. That is the strength of Pyrate actually. Line 246-248: I simply don’t understand what you mean, can you please explain what was NOT jointly estimated? Are some parameters collapsed into one or what? Line 252: why 20%? Seems like you discard 20% as a standard practice, but is it recommended? Shouldn’t you examine the convergence then discard with some cutoff after convergence has been reached? Line 258-260: I don’t understand what is done here. And what is “option -d” (rather only stating software input, you should explain what is being done as well, that is a general request). Line 269: “filtered six independent ones” filtered what? And how are they independent? Independent from what? I guess you “excluded” something (line 270), what proportion of species do you have left after excluding non-reef species? Were these also excluded from Fig. 1? If not, what are the (biological and sampling rate estimation) implications? Line 283, again sure, tell us which option you used (“lft”) but please tell us what this function performs. Line 284 “adjacent time samples” what does this mean? What are the time samples? Are they equal? What are the units of the “slope” you calculated? Aren’t these really net diversification rates? Why not just present the posterior of the diversification rates? It seems very odd, what is done here.

Along the same lines, in the paragraph starting with line 285, you use “divDyn” but I don’t know off the top of my head what divDyn does. Perhaps it is a function for per capita rates (line 291) based on which calculation? Is this Foote’s 2001 paper? Please cite the original paper and explain what is being done. divDyn is a R package but the methods are surely based on earlier papers? Or are there “new” approaches in divDyn? Line 288 any specific reason why 1 myr bins? PyRate disregards non-sampled species, Foote’s paper throws out also “singletons” these are “idiosyncrasies” of the approaches, but you should at least discuss what the assumptions of these approaches mean for your inferences (of diversity-dependence, for instance). The major peaks are the same, but it does not follow that your major conclusions on diversity-dependence will be the “same”. Maybe it is no point present estimate from divDyn if you don’t go further that just comparing peaks and troughs?

Line 312: “related species” related to what? Line 313 No, MCDD(MBD) was developed to look at diversity dependence NOT competition. More specifically, is there any indication that the species in question even use the same locations? Of course competition might lead to local exclusion, but these “competition concepts” are not testable with these data (or this MCDD model).

Fig. 3 Do I understand correctly that the HPD is relative to the black (all families) in 3c. to 3e.? I think it would be very helpful to show these for all the families as the arrows in 3a. and 3b. don’t give an indication of uncertainty.

Other points:

Title: “Fast-growing species shape the evolution of reef corals” I realize that the authors want to be general, but there is no presentation of data of the relative growth rates of the species in the different families compared, so it feels like this is stretching it. Why not just “acroporids shape the evolution of other reef corals”?

Line 51: I don’t understand the use of the word “therefore” here. What does “abundance” (in the previous sentence) logically have to do with competition? In addition, do you mean “abundance” as in

biomass or species richness? The latter is quantified but not “abundance”, in this ms.

Line 57 is also odd, the analyses is done on 50 million years, but this paragraphs claims that it is only In the Pleistocene acroporids had an effect? Again, the analyses don't quite line up with the writing. Line 64: I simply do not think you “reveal a unique mechanism”.....as you state here (as at least partially argued above).

I actually find it a bit odd that Fig. 1 b and d show what they show for acroporids but the underlying richness is as in Fig. 2 ... I would have expected a couple of blips in Fig 1b, and 1d. Maybe check your estimates/code?

Line 120: “given these results, we hypothesize...” The results being referred to I assume are on the previous page, where Acroporids increased in diversity in the Pleistocene. I do not understand how the hypothesis that it is “competition” follows.... Please explain. It's one thing to note that the analyses point that direction (if you want to interpret diversity-dependence as “competition”) but so what if Acroporid diversity increased in the Pleistocene? Should “competition” or “diversity-dependent speciation” “naturally” follow? Why?

Authors' replies to the comments by Reviewer #1

Comment#1 (R#1): This paper explores the importance of competition among reef corals over macroevolutionary timescales using broad fossil data and recently developed methods to assess diversity trajectories and their dependencies. The paper is particularly interesting because the role of competition is widely studied and well understood in ecological context, but its importance in shaping diversity patterns over evolutionary timescales have not been investigated before. Thus, this is significant and novel contribution to understand the evolution of reef systems and more generally, in understanding the role of biotic interactions in speciation-extinction dynamics.

Authors' reply: Thank you for recognizing the significance of our work, we were encouraged by your positive feedback. We addressed each of your concerns, which helped improving our manuscript. Please, find below a point-by-point response to each comment. Note that line numbers refer to the version with in-line track changes.

Comment#2 (R#1): The fossil data naturally are problematic. Current paper is based on extensive dataset, although it remains unclear what part of the full dataset (>24000 occurrences, >4000 species) were actually used in the final analyses (Cenozoic members of six selected families). Obviously full data were used to infer the preservation model. But it would still be good to know the number of species in the selected families and much they cover of the total standing reef coral diversity, especially because it is said that these families account for most of the occurrences in the dataset excluding extinct families.

Authors' reply: To comply with this suggestion, we have now expanded our methods section to include more information about the size of the family-level datasets used in the final analyses (lines 302-306). Please note that this information is also contained in Figure 3. We also included a sentence highlighting that the diversity within the six selected families covers approximately 40% of the total standing reef coral diversity (lines 301-302).

Comment#3 (R#1): This immediately raises the question of the role the now extinct families may have played. Presumably they were driven extinct by the competition, if the results obtained with extant species would hold, and this could certainly change the results to some degree. It would have been nice to explore them as well, or better justify their exclusion. Two independent methods

are used to estimate the diversification rates through time, and they seem to support similar conclusions.

Authors' reply: This is a great point, thanks for raising it. We agree that it would be nice to have an analysis including extinct families, however, our focus was in the coral families that form the bulk of present-day diversity on reefs (now stated on lines 105-110). Most of the extinct coral families that are abundant in the fossil record (e.g. Latomeandridae, Montlivaltiidae and Stylinidae) concentrate their occurrences and diversity in the Mesozoic, therefore, they have very little temporal overlap with extant families. This hinders the power of the model to detect any relevant effect of biological interactions (see Silvestro *et al.* 2017). To address your suggestion, we have now included a justification in the methods section (lines 303-304) that clarifies the reasoning behind the exclusion of extinct families.

Silvestro, D., Pires, M. M., Quental, T. B. & Salamin, N. Bayesian estimation of multiple clade competition from fossil data. *Evolutionary Ecology Research* 18, 41–59 (2017).

Comment#4 (R#1): The results are striking: the competitionally superior staghorn corals increase taxonomic turnover in other families, that is, both speciation and extinction. The authors are able to provide an explanation to this double-edged effect, by relating it to the capacity of staghorn corals to rapidly create complex reefs and hence, new niches for specialist species. I was wondering if it would be possible to backup this reasonable assumption with referencing some contemporary data?

Authors' reply: We have now included a reference to the empirical work by Tanner *et al.* (1994) that provides support to our assumption (lines 214-217). Within this long-term study of the scleractinian coral assemblage of Heron Island, Great Barrier Reef, the authors show that “poor competitors could replace superior ones indirectly by recruitment onto space recently vacated”. Given that free space seems to be constantly created by local disturbances on coral reefs (affecting acroporids disproportionately), it allows the persistence of opportunistic species within the system. This is the exact mechanism that we are proposing to have potentially enhanced species turnover rates through time.

Tanner, J. E., Hughes, T. P. & Connell, J. H. Species coexistence, keystone species, and succession: a sensitivity analysis. *Ecology* 75, 2204–2219 (1994).

Comment#5 (R#1): The methods are as sound as any available method can be in this admittedly somewhat speculative field of research. I also do not see any such flaws in the methodology and interpretation that could have been avoided by using any other approach.

Authors' reply: Thank you for supporting our methodological choices.

Authors' replies to the comments by Reviewer #2

Comment#1 (R#2): Siqueira et al. show compelling results on the importance of the diversification of a family of reef-building corals impacting speciation and extinction of other coral families. They combine the fossil record with speciation and extinction rate analyses, lineage through time plots, and cross-clade models to show the importance of the family Acroporidae in influencing the evolutionary trajectories of other reef-building coral families. In general, this paper is exciting, well-written, and the results and methods are robust. I just have some minor criticisms with interpretation of the results and would like to see these further addressed.

Authors' reply: Thank you for the positive comments, we appreciate the attention given to our work. We carefully addressed your criticisms and modified the manuscript accordingly. Our approach to addressing each comment is detailed in our responses below. Please, note that line numbers refer to the version with in-line track changes.

Comment#2 (R#2): How are you sure that it is competition is the driving force of extinction? You mention fast growth of acroporids likely promote their diversification, and this makes sense, but are there other processes that would promote the extinction/net-diversification of other reef building families? (i.e., environment is not conducive, other families are slow growing, etc).

Authors' reply: This is a very good point, thank you for bringing it up. Indeed, we cannot be sure that competition is the driving force of the effects of acroporids on the extinction rates

of other families. In fact, competition is difficult to be assessed even in contemporaneous ecological studies. However, given how widely prevalent competition for space is among present-day benthic organisms on coral reefs, we believe it to be the most plausible explanation for the extinction patterns we found. To reinforce our suggestion that diversity dependency is an important factor driving the diversity trajectories of coral families, we followed your recommendation and included three key environmental predictors (sea level, rate of sea-level change and paleotemperatures) to the MBD models (see Methods lines 377-387). In addition, we also added new analyses accounting for the location of the fossils to follow your suggestion in Comment #3. The results of these new analyses show that environmental predictors were generally less important than diversity-dependent mechanisms both in the global and in the regional datasets (see next response). This strengthens our initial findings and gives more support to the competition hypothesis.

Comment#3 (R#2): Also, it was not clear that you accounted for whether acroporidae co-occurred in the same location as other corals. If so, then I would be convinced that competition is important. Since you downloaded the fossil record database for your analyses, how did you account for whether the coral families co-occured in the same locales? In other words, how could Acroporidae in the Indo-Pacific outcompete Pocilloporidae in the tropical eastern Pacific?

Authors' reply: Another great point, thank you. In our global analyses, we did not account for whether the coral species co-occurred in the same locations. Unfortunately, the fossil record is sparse and does not allow such precise resolution to infer competition at site-specific occurrences. However, to comply with this suggestion, we have now performed the same family-level analysis, but limiting it to occurrences in the Indo-Pacific only (see Methods lines 420-422). In addition, we also performed an analysis selecting only collections in which the Acroporidae co-occurs with other families (see Methods lines 422-423). By filtering Indo-Pacific occurrences and co-occurring sites, the sample size per family gets too small to detect diversity dependent effects in family pairwise analyses, therefore, we were only able to perform these analyses using all families combined as a response variable. This dataset, with all families combined, is the best resolution that can be achieved while still retaining enough occurrences to examine diversity-dependency. Once again, the results were very similar

when compared to the initial findings (see Supplementary Figs. 2 and 3), suggesting that the Acroporidae might have indeed influenced overall rates of speciation and extinction in reef-building corals. This reinforces diversity-dependency (including competition) as an important mechanism driving coral macroevolutionary patterns. Finally, it must also be noted that our focus was on the Cenozoic and, for most of this time, there was global connectivity. Indeed, the hotspot of shallow-water marine biodiversity has moved halfway across the globe (Renema *et al.* 2008), so overlap of most taxa throughout this time is highly likely. The best way to capture this dynamic biogeographic history of reefs through time is by analysing the global dataset like we did in the main analyses.

Renema, W. *et al.* Hopping hotspots: global shifts in marine biodiversity. *Science* 321, 654–657 (2008).

Comment#4 (R#2): The cross clade network analyses are really cool. However, it is not clear how one family can simultaneously influence the speciation and extinction of another family. For example, in Figure 3a, Acroporidae strongly promotes speciation of Pocilloporidae and you show in Fig 3b that it also strongly promotes the extinction of the same family. Your explanation in the text on lines 170-192 is interesting, and I understand where Acroporidae might impact speciation of one family and extinction of another, but I don't understand how it could influence both in one family. Unless it did this at different times? Also, could you run these analyses on net diversification rates? Perhaps that would provide additional insight.

Authors' reply: There are two ways in which the cross-clade analysis can identify Acroporidae promoting both speciation and extinction in other families. First, you are correct in thinking that effects can vary with time. The MBD model detects correlations through time, however, the resulting effects represent an average estimated for the entire duration of the temporal overlap between lineages. Thus, the Acroporidae could have promoted both speciation and extinction in Pocilloporidae, for example, because these seemingly contrasting effects have been averaged across different times throughout their evolution. Second, the MBD model correlates the predictors (*i.e.*, diversity trajectories of other families and environmental factors) with the individual times of origination (T_s) and

extinction (T_e) of all species within a family. This means that increases in the number of lineages in Acroporidae at a specific time frame could be correlated with the extinction of some species, but also the origination of other species within Pocilloporidae, for example. Therefore, even within the same time period, one family can promote both speciation and extinction considering the responses of different lineages within the same affected family. We have now expanded our discussion to clarify these aspects of our results (lines 198-201). Unfortunately, the MBD analysis cannot be run on net diversification rates because it uses diversity trajectories as input rather than the rates themselves.

Comment#5 (R#2): Overall, nice work and a great read.

Authors' reply: Thank you for your constructive and positive comments.

Authors' replies to the comments by Reviewer #3

Comment#1 (R#3): Review for Fast-growing species shape the evolution of reef corals. In this contribution, the authors use data from the paleobiology to (lines 61-64): a) Described the major changes that determined the formation of present-day reef coral assemblages; b) Estimate the effects of diversity dependency across reef-building coral families and; c) Determine the relative importance of competition throughout the evolution of these families. In the abstract they conclude that (lines 24-25) “reef coral biodiversity experienced marked evolution rates shifts in the last three million years, likely driven by interspecific competition.”. Much as I enjoyed learning something about corals from this ms, I do not think that the authors have shown c) or can draw their conclusion stated in the paragraph above (copied from their abstract). I think a) needs some revision and b) is conditioned on assumptions that need to be clearer (see below).

Authors' reply: Thank you for your thorough review. We worked extensively to address your concerns, which helped making our work more robust. Please, find our detailed responses to each comment below. Note that line numbers refer to the version with in-line track changes.

Comment#2 (R#3): I will try to explain why in the following. In addition, I will try to clear on where the authors need to clarify their methods. Why are the conclusions problematic: Basically, the authors used PyRate to estimate speciation and extinction rates from fossil data (Fig. 1), and the consequence of these rates (Fig. 2) i.e. the number of lineages given these rates (estimated within the same Bayesian model framework PyRate). These estimates show how coral richness changed over geologic time (given the PyRate framework) but I don't see how what the patterns estimated say anything about "coral assemblages" in a) which are more or less local (or at best regional) communities. The authors have estimated time-varying rates for corals (in their different groupings) but they have not gone beyond that as a) seems to indicate. Perhaps it is just a poor choice of words.

Authors' reply: This is a very good point, thank you. Indeed, in the first version of the manuscript we used the word "assemblages" to represent the global dominance between families in terms of species richness. We agree that this was not the best choice of words, therefore, we reworded our objective *a* to clarify our introduction. It now reads: "(a) describe the major temporal changes in evolutionary rates that determined the formation of present-day species richness patterns among scleractinian coral families". We also modified every instance of the use of the word "assemblages" throughout the manuscript to better align with the objective rewording.

Comment#3 (R#3): While using the MBD/MCDD framework, which basically models richness in one group to be linearly correlated with speciation and extinction rates of itself or with other groups, "diversity-dependence" is explored, but this analysis hardly allows one to conclude that it is ecological competition that leads to consequences in speciation and extinction rates. At best one might conclude that there is some signal between richness and rates (i.e. diversity-dependence is detected), but that is all.

Authors' reply: To comply with this suggestion, we have now toned down the role of competition in our objectives to discuss diversity-dependent mechanisms more broadly. We agree that objective *c* in the original version was too focused on competition and, therefore, we removed it from the revised manuscript (lines 68-69). In addition, we rephrased the

suggested sentence of the abstract that now reads “By analysing the rich fossil record of Scleractinia, here we show that reef coral biodiversity experienced marked evolutionary rate shifts in the last 3 million years, likely driven by *biotic interactions*”. Finally, we changed many instances of the manuscript in which we had used the word *competition* to talk about *diversity dependency* instead (lines 60-61; 85-86; 91-93). As stated in our response to comment #2 by Reviewer #2, we still believe competition to be the most plausible explanation for the extinction patterns, given how prevalent it is among benthic organisms on present-day coral reefs. However, we agree that our analyses cannot identify ecological competition, so we now limit ourselves to talk about this potential mechanism in the discussion only.

Comment#4 (R#3): This is in part because, as the authors write in the paragraph starting with line 170, that there could be sea-level changes and other abiotic disturbances that could influence evolution. As an example, if sea-level was actually the primary driver of coral speciation, it could be the true underlying driver i.e. the solid arrows in Fig.3a, but because it was not modeled (not included in the analysis framework), one would conclude that it is Acroporidae that is driving speciation rates overall.

Authors’ reply: Thank you for raising this point. In response to this comment and to comment #2 by Reviewer #2, we have now included three key environmental predictors (sea level, rate of sea-level change and paleotemperatures) to the MBD models (see Methods lines 377-387). By doing this we are now able to show that biotic factors were generally more strongly correlated with evolutionary rates in reef-building corals than environmental factors (see Results and Supplementary Fig. 1). More interestingly, we found that the rate of sea level change is slightly correlated with speciation in Acroporidae, but not in other families (see Supplementary Fig. 1). This suggests that, indeed, sea level changes might have been an important driver of diversification in Acroporidae, which in turn affected other families through the mechanisms we had originally proposed. Therefore, these new results strengthen our initial findings and give more support to diversity dependency driven by Acroporidae as an important mechanism driving the evolution of reef corals.

Comment#5 (R#3): Without belaboring more specific examples, this means that how this ms is “sold” must be changed to reflect the underlying limitations of the analysis framework and data. If the authors really want to answer the questions they posed in the introduction, then other analyses must be carried out and other types of data collected.

Authors’ reply: As per response to comment #3, we have now reframed the objectives of our study to better reflect the reality of what can be inferred from the analyses. In addition, we performed new analyses (see response to comment #4) that strengthened our original results. We believe this has substantially improved the structure of our manuscript, thank you.

Comment#6 (R#3): In terms of the methods and description, I find it in general a bit wanting. E.g. the authors said they used the “complete fossil dataset” (line 216) to have “better estimates”. I don’t see any justification why this is so (nor have I come across that in the Pyrate literature).

Authors’ reply: Thank you for pointing this out. We have now removed the word “better” from the suggested sentence (lines 242-245) because it was indeed misplaced. We meant to say that we used the complete fossil dataset to permit diversification estimates throughout the whole timespan of scleractinian coral evolution.

Comment#7 (R#3): In Lines 224-225, be careful to say that Pyrate estimates are conditioned on sampled lineages (i.e. all the estimates are conditioned on sampling at least once, which means that there will be some bias).

Authors’ reply: We have now rephrased the suggested sentence to highlight this inherent bias of the fossil record (lines 257-260).

Comment#8 (R#3): In line 229 you cited Raup and Sepkoski for “commonly used” --- I am not sure if most people follow their estimation anymore? Please clarify. I would imagine Alroy (3 timer etc) or Foote (boundary crossers) are most commonly used?

Authors’ reply: Thanks for spotting this mistake, the citation number was incorrect. We meant to cite Silvestro *et al.* (2019) in that sentence, because this paper compared the

performance of PyRate against other commonly used methods (including Alroy’s three-timer and Foote’s boundary-crossing). We have also double checked the citation numbers throughout the manuscript to make sure they are all correct.

Silvestro, D., Salamin, N., Antonelli, A. & Meyer, X. Improved estimation of macroevolutionary rates from fossil data using a Bayesian framework. *Paleobiology* 45, 576–570 (2019).

Comment#9 (R#3): Line 232, the point of including extants is to have those fossil geologic ranges extend to the Recent right? Write that please if so, I think it is a bit misleading for people who don’t know how PyRate works.

Authors’ reply: We have added a new sentence to include this information (lines 262-263).

Comment#10 (R#3): Line 236, why 11 subsets and how did you do it (“randomly”? but how exactly?) Why 11? Are your results sensitive to how this “division” was done? Seems completely arbitrary?

Authors’ reply: The reason why we divided the dataset into eleven subsets is because we wanted to keep an equal number of species in each subset to facilitate model convergence. Because we had 4,235 species in total, eleven was the number that could divide our dataset into smaller and more manageable subsets that each had an equal number of species. We have now modified the referred sentence to make this point clearer (lines 265-267). As explained in lines 268-270, this was done to avoid convergence issues, given the large size of our original dataset and the consequent complexity of the model. The species kept in each subset were chosen randomly (now explained in lines 267-268), however, the final results are not sensitive to how this division was done. This is because we used this strategy as an intermediate step to obtain estimates of times of origination (T_s) and extinction (T_e) for each fossil lineage, which were subsequently combined to estimate the overall rates (λ and μ) through time (explained in lines 277-280). According to Silvestro *et al.* (2014), T_s and T_e are estimated on a lineage-specific basis and are based on the number of occurrences per lineage

per million years derived from the data. Therefore, as long as all occurrences from the same species are kept in the same subset (which they were), the estimates of T_s and T_e should be consistent and independent of the rest of the species in a certain subset.

Silvestro, D., Salamin, N. & Schnitzler, J. PyRate: a new program to estimate speciation and extinction rates from incomplete fossil data. *Methods in Ecology and Evolution* 5, 1126–1131 (2014).

Comment#11 (R#3): Line 239: 10 replicates is much less than the 100 recommended in PyRate. I think that it is important to account for age uncertainty, so I could recommend at more replicates here. That is the strength of Pyrate actually.

Authors' reply: We agree about the importance of accounting for age uncertainty. Thus, we followed your suggestion and increased the number of age resampling replicates to 50 (see Methods). The reason why we had initially run only 10 replicates was because each extra replicate adds a substantial amount of computational time, therefore, we considered 10 to be a good balance between efficiency and the amount of information gained from having more replicates. As you can see in the new figures, the overall results remained unchanged.

Comment#12 (R#3): Line 246-248: I simply don't understand what you mean, can you please explain what was NOT jointly estimated? Are some parameters collapsed into one or what?

Authors' reply: We have now modified the sentence to clarify this point (lines 277-280). What we meant to say in this sentence was that, rather than jointly estimating all parameters (T_s , T_e , λ and μ) from the full dataset (as in the original implementation of PyRate), we first focused on estimating T_s and T_e , so we could subsequently estimate overall λ and μ in a separate analysis (explained in lines 289-291). Originally, PyRate jointly estimates the times of origination (T_s) and extinction (T_e) for each fossil lineage and the rates of speciation (λ) and extinction (μ) for all lineages combined through time (explained in lines 251-254). However, because we had such a large dataset, reaching convergence with all these parameters was difficult. Therefore, we split the dataset into eleven subsets (as explained in

the response to comment #10) from which we estimated T_s and T_e only. These estimates were then combined across subsets (explained in lines 286-287) and used as input for another run of PyRate (option $-d$ – where T_s and T_e are fixed; see response to comment #14) to estimate overall λ and μ through time. This approach allowed us to achieve convergence in all parameters of interest.

Comment#13 (R#3): Line 252: why 20%? Seems like you discard 20% as a standard practice, but is it recommended? Shouldn't you examine the convergence then discard with some cutoff after convergence has been reached?

Authors' reply: Indeed, parameters should be checked after each run and the burn-in should be discarded after convergence was reached. However, given the sheer number of models we are running (50 rjMCMC replicates across 11 subsets [550 models], plus 50 rjMCMC replicates across 6 families [300 models], plus 50 MBD replicates across 6 families [300 models]), checking and defining specific convergence points for all parameters becomes unfeasible. Therefore, we used 20% burn-in as a conservative proportion of the posterior samples (10% is more commonly used) at which parameter convergence has been reached. In all models inspected visually, convergence was attained before getting to 20% of the samples. Importantly, changing this cut-off is very unlikely to have any effect on the final results, given that most of our parameters of interest are already being averaged across so many model replicates.

Comment#14 (R#3): Line 258-260: I don't understand what is done here. And what is "option $-d$ " (rather than only stating software input, you should explain what is being done as well, that is a general request).

Authors' reply: This has now been clarified within the text (lines 291-292). The option $-d$ in PyRate runs a model in which the times of origination (T_s) and extinction (T_e) for all fossil lineages are given as fixed values and, therefore, are not estimated. The only parameters estimated by this model are overall rates of speciation (λ) and extinction (μ). Using this

approach reduces model complexity (explained in lines 280-281) and facilitates parameter convergence (as explained in the response to comment #12).

Comment#15 (R#3): Line 269: “filtered six independent ones” filtered what? And how are they independent? Independent from what? I guess you “excluded” something (line 270), what proportion of species do you have left after excluding non-reef species? Were these also excluded from Fig. 1? If not, what are the (biological and sampling rate estimation) implications?

Authors’ reply: The six independent datasets resulted from selecting species corresponding to each of the selected reef-building coral families (Acroporidae, Agariciidae, Merulinidae, Mussidae, Pocilloporidae and Poritidae). We have restructured the sentence to clarify this point (lines 306-308). They are independent because each individual family dataset was used in separate PyRate models to estimate their diversity trajectories through time (Fig. 2). As explained in line 304, we did exclude species that are not classified as reef-associated because we were specifically interested in these environments (where interactions are most likely) at the family-level analysis. To clarify our methods, we have now included in lines 310-311 the information that the vast majority of fossil species within these families are reef-associated (Acroporidae – 83%; Agariciidae – 83%; Merulinidae – 85%; Mussidae – 77%; Pocilloporidae – 86%; Poritidae – 86%). In the case of Figure 1, we did not exclude non-reef-associated species because we were initially interested in exploring the diversification dynamics across all scleractinian coral fossils. Once again, the biological and sampling rate estimation implications are negligible, given that the majority of species (~70%) in the full scleractinian dataset are reef-associated.

Comment#16 (R#3): Line 283, again sure, tell us which option you used (“lt”) but please tell us what this function performs.

Authors’ reply: This information is now provided (lines 321-323).

Comment#17 (R#3): Line 284 “adjacent time samples” what does this mean? What are the time samples? Are they equal? What are the units of the “slope” you calculated? Aren’t these really net

diversification rates? Why not just present the posterior of the diversification rates? It seems very odd, what is done here.

Authors' reply: We have now clarified all the points raised within the manuscript (lines 323-326; 125-127). The time samples correspond to the diversity estimated at every 0.1 million years (derived from the option *-litt* in PyRate). From these time samples, we calculated the mean difference in diversity between subsequent samples backwards in time (i.e. diversity in time t was subtracted from diversity in time $t-1$), which gives a slope in number of species per Myr. Although this can also be interpreted as a rate, it is slightly different from diversification (as shown in Fig. 1) because the estimates of PyRate are more focused on detecting rate shifts, while Fig. 2b shows a more nuanced picture of how the mean diversity trajectories have changed through time.

Comment#18 (R#3): Along the same lines, in the paragraph starting with line 285, you use “divDyn” but I don’t know off the top of my head what divDyn does. Perhaps it is a function for per capita rates (line 291) based on which calculation? Is this Foote’s 2001 paper? Please cite the original paper and explain what is being done. divDyn is a R package but the methods are surely based on earlier papers? Or are there “new” approaches in divDyn? Line 288 any specific reason why 1 myr bins? PyRate disregards non-sampled species, Foote’s paper throws out also “singletons” these are “idiosyncrasies” of the approaches, but you should at least discuss what the assumptions of these approaches mean for your inferences (of diversity-dependence, for instance). The major peaks are the same, but it does not follow that your major conclusions on diversity-dependence will be the “same”. Maybe it is no point present estimate from divDyn if you don’t go further that just comparing peaks and troughs?

Authors' reply: Thank you for this suggestion. We have now clarified what the divDyn package does (lines 327-329), which is all based on previously published methods. We have also included the citation for the original paper that proposed the per capita rates used within the divDyn framework (line 339). Additionally, we provide more details about the differences between the approaches we used (lines 330-335). Please note that we did not claim that our major conclusions on diversity dependency are the same with the divDyn results. We only used divDyn as an alternative for estimating evolutionary rates and diversity trajectories

(now clarified in lines 332-334), and not for the subsequent models of diversity dependency (which can only be done through PyRate). This was important to assess the robustness of the macroevolutionary trends shown in Figs. 1 and 2 using alternative methods, but the differences between approaches have no consequences for our diversity dependency results.

Comment#19 (R#3): Line 312: “related species” related to what?

Authors’ reply: This sentence has now been rectified (lines 357-360). By “related” we meant other clades that could have experienced diversity-dependent effects.

Comment#20 (R#3): Line 313 No, MCDD(MBD) was developed to look at diversity dependence NOT competition. More specifically, is there any indication that the species in question even use the same locations? Of course competition might lead to local exclusion, but these “competition concepts” are not testable with these data (or this MCDD model).

Authors’ reply: We have changed the referred sentence to highlight that the MCDD model was originally developed to assess the effects of *negative* interactions (line 361). Additionally, as explained in the response to comment #3 by Reviewer #2, the coral fossil record is sparse and does not have enough resolution to allow us to assess competition at site-specific occurrences. To provide a more spatially focussed evaluation, we have now performed the same family-level analysis, but limiting it to occurrences from the Indo-Pacific only (see Methods lines 420-422). In addition, we also performed an analysis selecting only collections in which the Acroporidae co-occurs with other families (see Methods lines 422-423). By filtering Indo-Pacific occurrences and co-occurring sites, the sample size per family gets too small to detect diversity dependent effects in family pairwise analyses, therefore, we were only able to perform these analyses using all families combined as a response variable. This dataset, with all families combined, is the best resolution that can be achieved while still retaining enough occurrences to infer diversity-dependency. The results from these new analyses (see Supplementary Figs. 2 and 3) were similar to our initial findings, suggesting that Acroporidae species might have indeed influenced overall rates of speciation and extinction in reef-building corals. This, once again, highlights diversity-dependency

(including competition) as an important mechanism driving coral macroevolutionary patterns. Finally, it must also be noted that our focus was on the Cenozoic and, for most of this time, there was global connectivity. Indeed, the hotspot of shallow-water marine biodiversity has moved halfway across the globe (Renema *et al.* 2008), so overlap of most taxa throughout this time is highly likely. The best way to capture this dynamic biogeographic history of reefs through time is by analysing the global dataset like we did in the main analyses.

Renema, W. *et al.* Hopping hotspots: global shifts in marine biodiversity. *Science* 321, 654–657 (2008).

Comment#21 (R#3): Fig. 3 Do I understand correctly that the HPD is relative to the black (all families) in 3c. to 3e.? I think it would be very helpful to show these for all the families as the arrows in 3a. and 3b. don't give an indication of uncertainty.

Authors' reply: You are correct, the HPD intervals in Figures 3c to 3e correspond to the node shown in black in 3a and b. Following your suggestion, we now provide a new supplementary figure (6) that shows the uncertainty for all the family pairwise analyses.

Comment#22 (R#3): Other points: Title: “Fast-growing species shape the evolution of reef corals” I realize that the authors want to be general, but there is no presentation of data of the relative growth rates of the species in the different families compared, so it feels like this is stretching it. Why not just “acroporids shape the evolution of other reef corals”?

Authors' reply: Thank you for the suggestion. Indeed, we do not present any data on the relative growth rates of families, however, it is very well-established that species in Acroporidae are the fastest growing corals (see review by Pratchett *et al.* 2015) and comparing growth rates was not in the scope of the present study. Hence, given that fast-growth hints at the mechanisms we are proposing to explain acroporids promoting speciation and extinction rates in other coral groups, we opted for keeping our original title.

Pratchett, M. S. *et al.* Spatial, temporal and taxonomic variation in coral growth - implications for the structure and function of coral reef ecosystems. *Oceanography and Marine Biology: An annual review* 53, 215–295 (2015).

Comment#23 (R#3): Line 51: I don't understand the use of the word "therefore" here. What does "abundance" (in the previous sentence) logically have to do with competition? In addition, do you mean "abundance" as in biomass or species richness? The latter is quantified but not "abundance", in this ms.

Authors' reply: We agree that the word 'therefore' was unnecessary in that sentence, so we removed it. We also provide a definition for abundance in the Methods section (lines 298-301), which is the number of colonies per area. As abundance is correlated with the area occupied by colonies (Dietzel *et al.* 2021), it makes sense that the most abundant corals would compete more for space. We also agree that we did not quantify abundance in our manuscript, however, in the referred paragraph we were just making the case that interspecific competition is a widespread ecological phenomenon on present-day coral reefs.

Dietzel, A., Bode, M., Connolly, S. R. & Hughes, T. P. The population sizes and global extinction risk of reef-building coral species at biogeographic scales. *Nature Ecology and Evolution* 5, 663–669 (2021).

Comment#24 (R#3): Line 57 is also odd, the analyses is done on 50 million years, but this paragraphs claims that it is only In the Pleistocene acroporids had an effect? Again, the analyses don't quite line up with the writing.

Authors' reply: Thank you for raising this point. We agree that the text was not lining up with our analyses in that case. Thus, we modified the referred sentence that now reads: "Here we show that the diversification of Acroporidae (commonly known as staghorn corals) is associated with a major disruption in coral evolutionary patterns, suggesting strong diversity-dependent effects."

Comment#25 (R#3): Line 64: I simply do not think you “reveal a unique mechanism”.....as you state here (as at least partially argued above).

Authors’ reply: We appreciate your comment. However, given that all new analyses performed reinforce the pattern that we had originally described, we do believe we are revealing a novel mechanism. As far as we are aware, there has been no demonstration in the literature of a single taxonomic group that simultaneously enhances speciation and extinction in other ecologically similar groups. Therefore, we opted for maintaining the referred sentence in its original form.

Comment#26 (R#3): I actually find it a bit odd that Fig. 1 b and d show what they show for acroproids but the underlying richness is as in Fig. 2 ... I would have expected a couple of blips in Fig 1b, and 1d. Maybe check your estimates/code?

Authors’ reply: We have double checked our code and rerun all the analyses. The results remain unchanged. This is probably because PyRate focusses on detecting rate shifts through time, and given the intensity of the recent rate shift in Acroporidae, it might mask other smaller shifts that are apparent in the diversity trajectories.

Comment#27 (R#3): Line 120: “given these results, we hypothesize....” The results being referred to I assume are on the previous page, where Acroporids increased in diversity in the Pleistocene. I do not understand how the hypothesis that it is “competition” follows.... Please explain. It’s one thing to note that the analyses point that direction (if you want to interpret diversity-dependence as “competition”) but so what if Acroporid diversity increased in the Pleistocene? Should “competition” or “diversity-dependent speciation” “naturally” follow? Why?

Authors’ reply: Great point, thank you. We have now reformulated the paragraph (lines 129-140) to include the results of the new analyses accounting for environmental variables (see response to comment #4). After this reformulation, the referred sentence was removed and the connection between paragraphs is much clearer now.

REVIEWER COMMENTS

Reviewer #1 (Remarks to the Author):

The authors have taken all comments into account. I wish to congratulate the authors for good work and have no further suggestions.

Reviewer #2 (Remarks to the Author):

This is an exceptional body of work. The authors have responded to my criticisms, and I do appreciate the efforts that went into adding extra analyses.

Only minor criticism, I suggest looking at the first few sentences of the abstract and modify accordingly. Currently, the first few sentences primarily focus on competition.

Great job.

Reviewer #4 (Remarks to the Author):

This is a very interesting study on the use of the Cenozoic fossil record to infer speciation and extinction rates in Scleractinia, and to explore their relationships within and between clades. I have been tasked to specifically assess the paper in the context of the issues raised by Reviewer 3, but having examined the paper in its entirety, I'm convinced that the authors have presented a robust study with cutting-edge and thoughtful analyses on a rich fossil dataset. The study sets a high standard for macroevolutionary studies on corals.

I am however concerned about the inferences drawn from the analyses. Importantly, only macroevolutionary patterns, over evolutionary timescales, can be inferred from this dataset. There are no ecological data presented, not even growth rates that are assumed for fast- vs. slow-growing corals. Indeed, at line 55: 'Yet, it is not known whether spatial competition or other diversity-dependent biological factors can scale up to determine large-scale macroevolutionary patterns on coral reefs.' This remains untested because there are neither spatial analyses nor test of diversity-dependence. Rather, the study assumes diversity-dependence in order to test for self- and cross-clade effects of diversification; there are no analyses relating diversification with diversity to get at a particular or general limit of diversity over time and/or space.

The focus on interspecific competition still persists (e.g. first line!) despite the previous reviewers' concerns. The most appropriate way to test for competition is to examine co-occurring corals in contemporaneous facies (e.g. <http://doi.org/10.1017/pab.2019.3>), and not throughout the entire clade. The fact that there is limited self-diversity dependency 'given that confamilial species generally share more similar ecologies' suggests that these ecological concepts may not hold at an evolutionary timescale, especially when there are no relevant data (i.e. contemporaneous fossils or even adjacent corals).

There have been many instances in the literature where such analyses have been presented inappropriately, leading to erroneous inferences. A famous case is the perceived competition between bivalves and brachiopods leading to the former replacing the latter right around the end of the Paleozoic. Competition is just really hard to distinguish in the fossil record. Today's reefs with their structural and topographical complexity attests to the real possibility that many coral species can coexist without really competing and eventually showing up as evolutionary/fossil signals.

Another major issue with the conclusion is that there is a wide range of growth rates in Acroporidae. Only branching Acropora has the effect of being fast-growing, with many other growth forms (and taxa) exhibiting slower growth. In fact, on average, families such as Pocilloporidae and

Psammocoridae have comparable growth rates (see e.g. Figure I in <https://doi.org/10.1016/j.tree.2016.02.012>), so attributing or relating the diversification effects to growth rates is problematic (especially when there are no data/analyses to support it). It could even be argued that diversification would be accelerated when closely-related species can limit similarities, so to suggest that the species contributing to high diversification all have fast growth rates run counter to the main conclusion that biological/ecological interactions shape macroevolutionary patterns.

Authors' replies to the comments by Reviewer #1

Comment#1 (R#1): The authors have taken all comments into account. I wish to congratulate the authors for good work and have no further suggestions.

Authors' reply: Thank you. We appreciate your time and effort in checking our responses to your initial comments.

Authors' replies to the comments by Reviewer #2

Comment#1 (R#2): This is an exceptional body of work. The authors have responded to my criticisms, and I do appreciate the efforts that went into adding extra analyses. Only minor criticism, I suggest looking at the first few sentences of the abstract and modify accordingly. Currently, the first few sentences primarily focus on competition. Great job.

Authors' reply: We are grateful for your compliments and pleased that you recognize our effort to address your initial concerns. In this new version, we modified the first sentences of the abstract to deal with your suggestion (Lines 18-23). Once again, thank you for your time and for the valuable comments that helped improve our manuscript.

Authors' replies to the comments by Reviewer #4

Comment#1 (R#4): This is a very interesting study on the use of the Cenozoic fossil record to infer speciation and extinction rates in Scleractinia, and to explore their relationships within and between clades. I have been tasked to specifically assess the paper in the context of the issues raised by Reviewer 3, but having examined the paper in its entirety, I'm convinced that the authors have presented a robust study with cutting-edge and thoughtful analyses on a rich fossil dataset. The study sets a high standard for macroevolutionary studies on corals.

Authors' reply: Thank you for your kind and positive comments. We are also grateful for your time and attention in checking our responses to Reviewer #3. Please find our detailed responses to each of your concerns below. Note that line numbers refer to the version with in-line track changes.

Comment#2 (R#4): I am however concerned about the inferences drawn from the analyses. Importantly, only macroevolutionary patterns, over evolutionary timescales, can be inferred from this dataset. There are no ecological data presented, not even growth rates that are assumed for fast- vs. slow-growing corals.

Authors' reply: Thank you for raising this point. We agree that our study did not originally include any ecological data that would allow us to make inferences about fast- vs. slow-growing corals. Therefore, to deal with this point we have now included a supplemental analysis of growth rates across the focus coral families (Supplementary Fig. 4). In this analysis, we downloaded growth data from the Coral Trait Database (<https://coraltraits.org>), which shows that acroporids are indeed the fastest growing coral family when compared to other families in our manuscript. This new data supports the use of rapid growth rates in acroporids as a potential explanatory mechanism for some of the macroevolutionary trends observed. Although we agree that not all growth forms in the Acroporidae grow faster than other coral families (*cf.* comment #6; Madin *et al.* 2016 – *Trends Ecol. Evol.*), the vast majority of species within the family are at the fast-growing end of the spectrum (Figure 3 in Roff 2021 – *BioScience*). These fast-growing species were also suggested to underpin the recent numerical dominance of acroporids in the fossil record (Renema *et al.* 2016 – *Sci. Adv.*). Hence, we feel obliged to address the suggestion that the fast growth of the bulk of acroporid species may have been associated with some of the results described in our manuscript. We do accept, however, that although it is possible, or even probable, we do not directly demonstrate this. We have, therefore, modified the text accordingly.

Comment#3 (R#4): Indeed, at line 55: ‘Yet, it is not known whether spatial competition or other diversity-dependent biological factors can scale up to determine large-scale macroevolutionary patterns on coral reefs.’ This remains untested because there are neither spatial analyses nor test

of diversity-dependence. Rather, the study assumes diversity-dependence in order to test for self- and cross-clade effects of diversification; there are no analyses relating diversification with diversity to get at a particular or general limit of diversity over time and/or space.

Authors' reply: We appreciate your comment and to comply with your suggestion, we have now modified the suggested sentence to read: “*Yet, it is not known whether competition for space or other biological interactions can scale up to determine large-scale macroevolutionary patterns on coral reefs*” (Lines 57-59). In this context, it is important to note that the models are specifically designed to test for diversity dependence. However, this is a specific type of diversity dependence that is defined on lines 62-63; *i.e.* “*when diversification in one lineage impacts the evolution of others*”. The models used in our manuscript were specifically designed to test for evidence of such an interactive effect (*e.g.* Silvestro *et al.* 2015 – *PNAS*). They do not demonstrate that interactions occurred, but that patterns of diversification point to the probability of such interactions. The statistical framework that we employed detects diversity correlations between both biotic and abiotic predictors through evolutionary time (as noted in your comment #2), allowing us to test the relative importance of each factor. Although a negative correlation between family diversity trajectories cannot be mechanistically linked to interspecific competition, we offer it as a potential explanation for our extinction rate results. Indeed, it may be the most parsimonious explanation, given how prevalent this interaction is on present-day coral reefs. We agree that the concept of diversity dependence sometimes encompasses the search for a general limit of diversity over space and time. However, given that the analyses specifically focus on the overall cross-lineage effects on diversification, unfortunately describing limits to biodiversity is beyond the scope of our analyses.

Comment#4 (R#4): The focus on interspecific competition still persists (*e.g.* first line!) despite the previous reviewers' concerns. The most appropriate way to test for competition is to examine co-occurring corals in contemporaneous facies (*e.g.* <http://doi.org/10.1017/pab.2019.3>), and not throughout the entire clade. The fact that there is limited self-diversity dependency ‘given that confamilial species generally share more similar ecologies’ suggests that these ecological concepts

may not hold at an evolutionary timescale, especially when there are no relevant data (i.e. contemporaneous fossils or even adjacent corals).

There have been many instances in the literature where such analyses have been presented inappropriately, leading to erroneous inferences. A famous case is the perceived competition between bivalves and brachiopods leading to the former replacing the latter right around the end of the Paleozoic. Competition is just really hard to distinguish in the fossil record. Today's reefs with their structural and topographical complexity attests to the real possibility that many coral species can coexist without really competing and eventually showing up as evolutionary/fossil signals.

Authors' reply: In response to this comment, we have now modified the abstract to reduce the focus on interspecific competition even further (Lines 18-23). As noted in our response to comment #3, we have endeavoured to separate the description of patterns, which we can do with some confidence, from the mechanistic basis which we only infer and not establish. Competition remains as a likely explanation for the extinction results. However, we keep this as a suggestion in the discussion, and for context in the introduction.

We agree that directly assessing competition in the fossil record is complex and would require different methods from the ones used in our manuscript. And it can probably never be demonstrated effectively based on fossils. However, in the case of reef-building corals, there has been a previous study that demonstrated how the proportional numerical dominance of acroporids is a product of the last three million years (Renema *et al.* 2016 – *Sci. Adv.*). The approach used by Renema *et al.* (2016) is similar to the one used in the bivalves vs. brachiopods example (Hsieh *et al.* 2019 – *Paleobiology*), and their results are in perfect alignment with our findings. We specifically avoided replicating the results of Renema *et al.* (2016) to prevent a potential overlap between papers and to provide independent, complementary, perspectives.

Comment#5 (R#4): Another major issue with the conclusion is that there is a wide range of growth rates in Acroporidae. Only branching *Acropora* has the effect of being fast-growing, with many other growth forms (and taxa) exhibiting slower growth. In fact, on average, families such as Pocilloporidae and Psammocoridae have comparable growth rates (see e.g. Figure I in

<https://doi.org/10.1016/j.tree.2016.02.012>), so attributing or relating the diversification effects to growth rates is problematic (especially when there are no data/analyses to support it). It could even be argued that diversification would be accelerated when closely-related species can limit similarities, so to suggest that the species contributing to high diversification all have fast growth rates run counter to the main conclusion that biological/ecological interactions shape macroevolutionary patterns.

Authors' reply: This is an insightful observation. To address this issue, we have now included a supplemental analysis of growth rates across coral families that highlights the fact that most species within Acroporidae indeed have faster growth rates (Supplementary Fig. 4). As noted in our response to comment #2, this new analysis also supports the retention of rapid growth in the Acroporidae as a possible explanation in the discussion. In the text, we note that “*The only family that enhanced overall speciation (Acroporidae; Fig. 3c) is mostly composed of fast-growing branching species that are major ecosystem engineers that contribute to habitat complexity*” (Lines 159-162); *i.e.* fast growth is a common feature of this family. We believe that the new analysis and restructured text clarify the attribution of the diversification effects to growth rates. Thank you for helping us improve this point in our manuscript.

REVIEWER COMMENTS

Reviewer #4 (Remarks to the Author):

I appreciate the authors' responses to my comments. This is an excellent study that has no doubt advanced our understanding of coral macroevolution.

Authors' replies to the comments by Reviewer #4

Comment#1 (R#4): I appreciate the authors' responses to my comments. This is an excellent study that has no doubt advanced our understanding of coral macroevolution.

Authors' reply: Thank you for your kind words. We are glad that you enjoyed reading our revised manuscript.